# Range-Bounded Adaptive Therapy in Metastatic Prostate Cancer

**DOI:** 10.3390/cancers14215319

**Published:** 2022-10-28

**Authors:** Renee Brady-Nicholls, Heiko Enderling

**Affiliations:** 1Department of Integrated Mathematical Oncology, H. Lee Moffitt Cancer Center and Research Institute, Tampa, FL 33612, USA; 2Department of Genitourinary Oncology, H. Lee Moffitt Cancer Center and Research Institute, Tampa, FL 33612, USA; 3Department of Radiation Oncology, H. Lee Moffitt Cancer Center and Research Institute, Tampa, FL 33612, USA

**Keywords:** adaptive therapy, competition, prostate cancer, mathematical modeling, patient-specific

## Abstract

**Simple Summary:**

The success of adaptive therapy (AT), whereby treatment is cycled on and off using patient-specific treatment triggers, is hypothesized to be due to the competitive suppression of the resistant population by the sensitive population. There may exist a subset of patients who might benefit from slight modifications to AT, whereby treatment is initially delayed and subsequently cycled to remain between predetermined, patient-specific bounds, i.e., range-bounded adaptive therapy (RBAT). Here, we investigate the potential benefit of RBAT using a previously calibrated and validated model of stem cell and prostate-specific antigen dynamics. Simulations show that RBAT can further extend time to progression, while reducing the cumulative dose received. By delaying treatment, competition is further leveraged between the sensitive and resistant populations and treatment response can be prolonged.

**Abstract:**

Adaptive therapy with abiraterone acetate (AA), whereby treatment is cycled on and off, has been presented as an alternative to continuous therapy for metastatic castration resistant prostate cancer (mCRPC). It is hypothesized that cycling through treatment allows sensitive cells to competitively suppress resistant cells, thereby increasing the amount of time that treatment is effective. It has been proposed that there exists a subset of patients for whom this competition can be enhanced through slight modifications. Here, we investigate how adaptive AA can be modified to extend time to progression using a simple mathematical model of stem cell, non-stem cell, and prostate-specific antigen (PSA) dynamics. The model is calibrated to longitudinal PSA data from 16 mCRPC patients undergoing adaptive AA in a pilot clinical study at Moffitt Cancer Center. Model parameters are then used to simulate range-bounded adaptive therapy (RBAT) whereby treatment is modulated to maintain PSA levels between pre-determined patient-specific bounds. Model simulations of RBAT are compared to the clinically applied adaptive therapy and show that RBAT can further extend time to progression, while reducing the cumulative dose patients received in 11/16 patients. Simulations also show that the cumulative dose can be reduced by up to 40% under RBAT. Through small modifications to the conventional adaptive therapy design, our study demonstrates that RBAT offers the opportunity to improve patient care, particularly in those patients who do not respond well to conventional adaptive therapy.

## 1. Introduction

The emergence of resistance presents a major challenge in the treatment of many cancer types, specifically in prostate cancer which remains the second most prevalent cancer and the fifth leading cause of cancer death among men [1]. Treatment is traditionally administered at maximum tolerated dose (MTD) with the intent to completely eradicate the cancer. Such an approach fails to consider the evolutionary dynamics of treatment response whereby competition between sensitive and resistant phenotypes contributes to therapy failure. Suppression of the resistant cells by the sensitive cells is inhibited during treatment, allowing for the competitive release of the resistant phenotype if treatment is administered for a prolonged period of time [2].

Intermittent or adaptive therapy, whereby treatment is cycled on and off based on pre-determined levels, has been shown to be an effective strategy to sufficiently leverage the competition between the sensitive and resistant populations [3,4,5,6,7,8,9,10]. Resistant cells are typically less fit than sensitive cells due to a cost of resistance allowing for sensitive cells to proliferate at the expense of the resistant cells. Adaptive therapy aims to maintain sufficient levels of sensitive cells during therapy such that resistant cell proliferation can be effectively suppressed during treatment holidays. While treatment holidays allow for tumor regrowth, the resistant cell population remains small so that retreatment remains effective.

While the competitive suppression of the resistant cell population should be an effective strategy to prolong response, there are cases where adaptive therapy fails. In these cases, often due to the infrequent clinical measurements, treatment might completely deplete the sensitive population thereby removing the resistant cell growth inhibition during a treatment holiday [11]. Alternatively, the initial sensitive cell population might not be sufficient to compete with the resistant cells, resulting in tumor growth due to further selection for the resistant population during treatment. Similarly, using adaptive therapy with a low initial resistant population may result in earlier progression than standard of care treatment at maximally tolerated dose, due to the introduction of de novo resistance [12]. To solve this problem, Hansen and Read proposed that patients may tolerate slight increase in tumor burden, and that withholding treatment initially could allow sufficient time for the sensitive population to grow, thereby enhancing the competition between the cell populations as well as the efficacy of adaptive therapy once treatment begins [13]. In the adaptive therapy design, treatment is continued until the burden falls below 50% of its pretreatment level and resumes once the burden rises above the pretreatment level again [3]. These successive 50% reductions limit the competitive suppression by reducing the average size of the sensitive population. In the modifications proposed by Hansen and Read, the burden is allowed to rise above the initial level by some proportion, called the acceptable baseline. Treatment commences once the burden exceeds the acceptable baseline by a measurable amount and continues until it falls below the acceptable baseline again. By maintaining the burden between these bounds, it is hypothesized that a competitive average size of sensitive population is maintained and therefore competition is enhanced [13].

Zhang et al. developed an evolutionary game theory model using Lotka-Volterra equations with three competing cancer populations to demonstrate how adaptive therapy, using patient-specific burden dynamics to inform on and off treatment cycles, successfully suppresses the resistant population, while reducing the cumulative drug dose when compared to continuous treatment at MTD [3]. This motivated a pilot clinical study (NCT02415621) in metastatic castration resistant prostate cancer (mCRPC) patients undergoing adaptive abiraterone acetate therapy [3,5,14]. We have previously developed a simple model of prostate-specific antigen (PSA) evolution dictated by prostate cancer stem cell dynamics that was calibrated to the longitudinal prostate-specific antigen (PSA) from 70 biochemically recurrent prostate cancer patients undergoing intermittent androgen deprivation therapy (ADT) [15] and 16 mCRPC patients undergoing adaptive AA in the pilot study [16]. We demonstrated how patient-specific evolutionary dynamics from individual treatment cycles from both biochemically recurrent and mCRPC patients could predict patient response with 89% and 81% accuracy, respectively. In this study, we will use our simple model to investigate how the proposed modifications from Hansen and Read could be used to enhance adaptive therapy in individual patients.

## 2. Materials and Methods

### 2.1. Clinical Data

Sixteen mCRPC patients undergoing adaptive therapy (AT) as part of the pilot clinical study at Moffitt Cancer Center (NCT02415621) were included in the study. Patients received abiraterone acetate plus prednisone as standard of care prior to trial registration. Those who achieved a 50% or more decline in their PSA were eligible for the trial. Patients received treatment until PSA declined to 50% of its pretreatment value. Treatment was resumed once PSA rose above the pretreatment level and continued until it fell below 50% of the new pretreatment level. PSA was monitored every 4 to 6 weeks, with restaging bone, pelvic, and abdominal scans performed every 12 weeks. Patients continued to cycle on and off of treatment until developing radiographic progression based on PCWG2 criteria. Additional details regarding patient demographics and data collection can be found in [3,5,14].

### 2.2. Mathematical Model

Our previously developed mathematical model of PSA, prostate cancer stem cell, and prostate cancer non-stem cell dynamics was calibrated to the longitudinal PSA data. Prostate cancer stem cells are resistant to AA, while prostate cancer non-stem cells are sensitive. Thus, they will, respectively, be referred to as resistant and sensitive hereafter. In the model, the resistant cells $\left(R\right)$ divide at rate λ [day^−1^] to produce either one resistant stem cell and one sensitive non-stem cell (asymmetric division) or two resistant (symmetric division) stem cells. This is modulated by the probability of symmetric vision parameter $p_{s}$. The sensitive cells $\left(S\right)$ produce PSA $\left(P\right)$ at rate ρ [μg/L day^−1^], and die at rate α [day^−1^] in response to treatment. PSA decays at rate φ [day^−1^] (Figure 1a).

Previous analysis has shown that λ, the rate at which the resistant cells divide can be approximated as once a day, i.e., $\lambda =\ln\left(2\right)$ day^−1^. Additionally, sensitivity and identifiability analysis showed that ρ and φ could be uniform across all patients, while $p_{s}$ and α be patient-specific without significantly changing the model results [15]. Parameter optimization, via a leave-one-out analysis, was previously used to find suitable parameter values such that the model is able to accurately describe PSA dynamics for each patient undergoing adaptive therapy [16].

### 2.3. Conventional Adaptive and Range-Bounded Adaptive Therapy

Though the patients were evaluated based radiographic progression, due to the nature of our model we compare conventional adaptive therapy (AT) and RBAT based on PSA progression. PSA progression is defined as PSA increasing ≥25% and at least 2 ng/mL above the nadir, confirmed by a second measurement three or more weeks later.

Ten of the 16 patients developed PSA progression prior to the conclusion of the trial. The optimal model parameters were used to simulate these patient responses during AT. For the remaining six patients who continued to respond after the trial completion date, the optimal parameter values were used to fit their PSA data up to the trial completion date and AT was simulated forward until PSA progression. That is, the pre-treatment PSA value was determined from the last cycle of available data and that was used to simulate on- and off-treatment cycles until progression (Figure 1b). The model was simulated until either PSA progression or until the end of the simulation at 10 years. For RBAT simulations, PSA is initially allowed to rise above an acceptable baseline β, defined as $\left(1+x\right)p_{1},$ where $p_{1}$ is the initial PSA and 0≤x≤1. Treatment is turned on once PSA rises above a treatment trigger τ, defined as $\left(1+y\right)\beta$ for 0≤y≤1, and continues until PSA falls below β. Treatment was cycled off and on until PSA progression (Figure 1c). The RBAT range-bound for PSA is between β and τ.

## 3. Results

Range-Bounded Adaptive Therapy Prolongs Treatment Response

Model simulations of AT and RBAT were compared for each patient. Simulations show that RBAT can extend TTP while decreasing the cumulative dose received. In particular, Patient 1009 continued to respond to AT after 20 months of trial follow up, undergoing two cycles of therapy. The model was simulated forward and found that he would progress under AT during the fourth cycle at 60 months. RBAT simulations showed that progression could be extended by an additional 15+ months. In this case, PSA was allowed to increase to 25% above the initial PSA value (acceptable baseline $\beta =1.25p_{1}$) and treatment began once PSA rose 50% above the acceptable baseline (treatment trigger τ=1.50β). Treatment was turned off once PSA fell below β and continued to cycle on and off until PSA progression. The cumulative dose received was also reduced by 18% (Figure 2a). Patient 1005 progressed under AT after 21 months of treatment, However, simulations showed that RBAT could extend TTP to almost 40 months, while reducing cumulative dose by 20% (Figure 2b).

RBAT was simulated using acceptable baseline values of 10%, 25%, and 50% above the initial PSA value and treatment trigger values of 10%, 50%, and 100% above the acceptable baseline. Simulation comparisons for all patients showed that RBAT was able to extend TTP for 11/16 patients (69%) and reduce cumulative dose received for 15/16 patients (94%; Figure 3a). Simulations showed that four patients (Patients 1004, 1011, 1012 and 1017) would continue to respond to both AT and RBAT for over 10 years. Model simulations for Patient 1002 showed that he would respond to AT for more than 10 years. RBAT simulations showed that progression would occur at between 32 and 44 months, depending on the chosen β and τ values. The cumulative dose reduction was also dependent on the $\left(\beta ,\tau \right)$ pair. In general, the cumulative dose received was reduced and TTP was extended with RBAT compared to AT (Figure 3b). Kaplan–Meier analysis showed that the median time to PSA progression was also significantly extended with RBAT (Figure 3c–e) for all $\left(\beta ,\tau \right)$ combinations.

## 4. Discussion

In this study, we have used a simple mathematical model of PSA dynamics to investigate how adaptive therapy can be further modified based on ecological and evolutionary principles [2,6,8,13] to further improve patient response. The model was previously calibrated and validated against longitudinal PSA data from 16 mCRPC patients undergoing AT. Model parameters were used to simulate a novel concept of range-bounded adaptive therapy, or RBAT, and to compare patient response dynamics between AT and RBAT.

Acceptable baseline values ranging from 10–50% above the patient-specific initial PSA value and treatment triggers ranging from 10–100% of the acceptable baseline value were used. Our results show that RBAT can extended TTP for a large proportion of patients (11/16) for all combinations of β and τ, with just one patient predicted to have longer TTP with AT when compared to RBAT. Simulations also showed that the cumulative dose can be reduced by up to 40% under RBAT depending on the chosen (β, τ) pair.

Comparison of TTP and cumulative dose reduction showed that β can be confidently increased (Figure 3). That is, TTP is extended for all values of β, with τ having a minimal effect on how much TTP is increased (Figure 3c–e). It should be noted that to be in line with clinical practice, PSA values were checked every four weeks to determine PSA progression. Consequently, PSA was able to rise well above the given treatment trigger for some patients, resulting in non-significant differences in TTP between larger τ values for a given β. For instance, for Patient 1005 with $\beta =1.25p_{1}$, TTP differed by just 28 days for τ ranging between 10% and 100% above the acceptable baseline value. However, both β and τ had significant effects on the reduction in the cumulative dose for this particular patient.

For 4/16 patients (Patient 1004, 1011, 1012, and 1017) RBAT did not change TTP, as model simulations showed that they would continue to respond for more than 10 years under both AT and RBAT. However, the cumulative dose was reduced by 5–25% under RBAT for each of these patients (Figure 3a), demonstrating that RBAT can potentially improve quality of life in terms of less time on treatment. Patients 1003, 1007, and 1016 progressed under AT prior to the end of the trial. Model simulations showed that RBAT would extend TTP to more than 10 years and this was insensitive to the values for β and τ, though the change in the cumulative dose varied significantly depending on the chosen β and τ values.

A limitation of waiting for PSA to rise above τ during RBAT is that PSA might rise significantly above the given treatment trigger over a short period of time before treatment is resumed. Model simulations for Patient 1005 showed that PSA was able to remain bounded between β and τ for four cycles. However, during the last treatment holiday, PSA rose from 114 ng/mL to 382 ng/mL (an increase of more than 300%) over a period of two months (Figure 2b). This sharp increase significantly selected for the resistant cells. Model simulations showed that resuming treatment at this time would result in PSA progression (PSA began to rise during treatment). As these patients are metastatic, a three-fold increase in PSA could potentially have detrimental effects such as the development of new metastases, as well as other symptoms related to quality of life.

The same can be said regarding the initial waiting period prior to the commencement of treatment. The initial PSA values varied from 2.42 to 109 ng/mL for the patients included in this study. Though simulation results showed that increasing β and τ would result in the longest TTP for the majority of the patients, this might not be feasible for those with higher initial PSA values. New metastases, bone pain, a decline in physical ability, and greater morbidity can occur when treatment is withheld. Determining who is a good candidate for RBAT is dependent upon the individual patient and requires balancing the potential benefits of prolonged survival and reduced drug use with the potential risks. However, the wide range of initial PSA values for the trial patients supports the argument that PSA can be safely increased for some patients. These limitations are likely due to the small study sample size. The optimal (τ,β) pair is potentially dependent on additional patient-specific factors such as the number of metastases and a patient’s toxicity levels. Future analysis with a larger data set may provide more insight into how these factors can be incorporated to determine appropriate (τ,β) combinations.

For those patients who are good candidates for RBAT, our model could potentially be used as a tool to determine when PSA will rise above a given treatment threshold such that treatment begins as soon as possible. The model could also be used to propose treatment adaptations after the first cycle of AT once sensitive and resistant cell dynamics have been estimated.

## 5. Conclusions

Here, we focused on a calibrated and validated PCa stem cell model to test RBAT. Many mathematical models have been developed for prostate cancer dynamics and AA based on different biological mechanisms of evolution of resistance to hormone therapy [17,18,19,20]. Future work will include Bayesian analysis of the different mathematical models and biological mechanisms of development of resistance to ensure robustness of the presented results before recommendations for prospective clinical validation should be made [21,22].

## Figures and Tables

**Figure 1 cancers-14-05319-f001:**
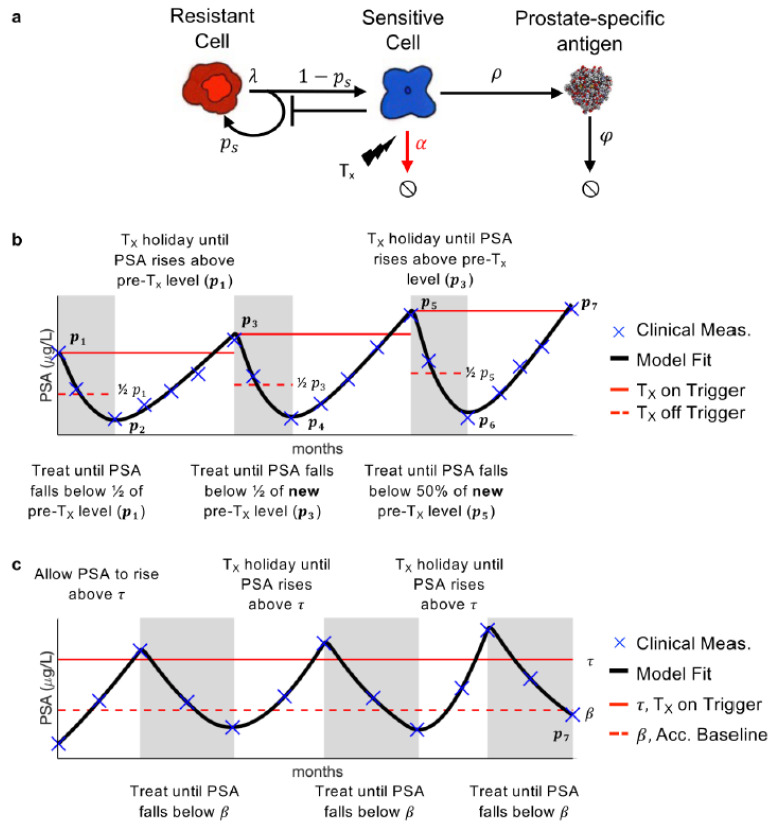
Model and treatment designs. (**a**) Model schematic of treatment resistance stem cells, sensitive non-stem cells, and prostate-specific antigen interactions. (**b**) Adaptive therapy treatment design. (**c**) Range-bounded adaptive therapy treatment design.

**Figure 2 cancers-14-05319-f002:**
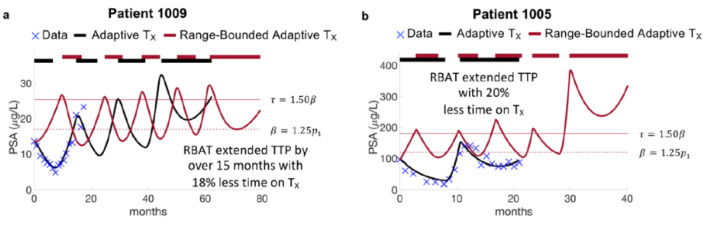
Model fits to adaptive therapy and simulations of range-bounded adaptive therapy. (**a**) Patient 1009 responded to adaptive until the end of the trial at 20 months. Model simulations forward showed that he would develop PSA progression after 60 months on adaptive therapy (black curve). Model simulations of range-bounded adaptive therapy show that time to progression can be extended by over 15 months with reduced cumulative dose. (**b**) Patient 1005 progressed on the trial after 20 months. Model simulations of RBAT extended time to progression to 40 months.

**Figure 3 cancers-14-05319-f003:**
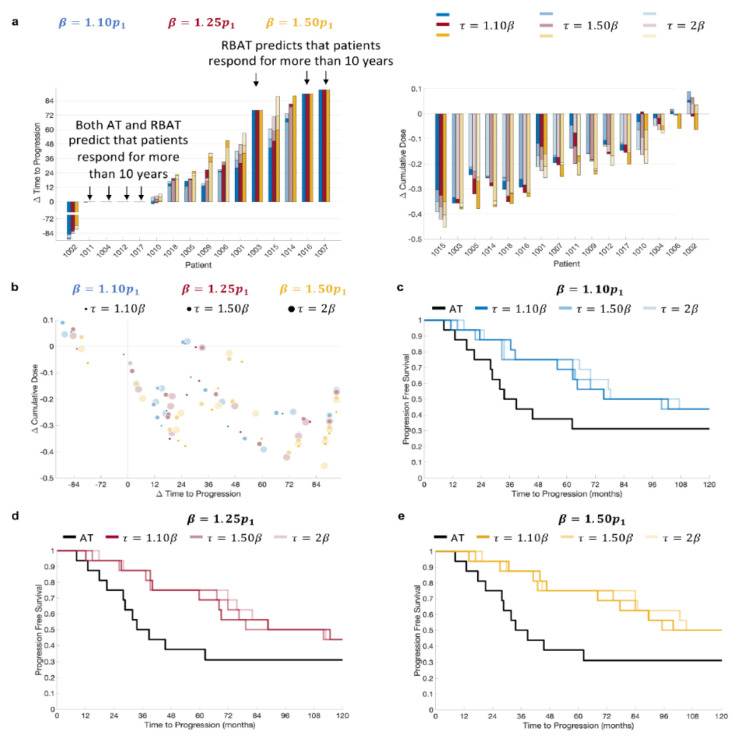
Model simulation change in time to progression and cumulative dose comparisons. (**a**) Comparison of patient-specific (left) change in time to progression and (right) change in cumulative dose for $\beta =\left(1.10p_{1},1.25p_{1},\;1.50p_{1}\right)$ and $\tau =\left(1.10\beta ,\;1.50\beta ,\;2\beta \right)$. (**b**) Change in time to progression versus change in cumulative dose. Kaplan–Meier plots for (**c**) $\beta =1.10p_{1}$, (**d**) $\beta =1.25p_{1}$, and (**e**) $\beta =1.50p_{1}$ with $\tau =\left(1.10\beta ,\;1.50\beta ,\;2\beta \right)$.

## Data Availability

The data supporting the findings of this article are available upon reasonable request of the corresponding author.

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
