# Peer review of "Range-Bounded Adaptive Therapy in Metastatic Prostate Cancer"

_cancers, 2022, doi:10.3390/cancers14215319_

Round 1

Reviewer 1 Report

The present research article entitled “Range-bounded adaptive therapy to further leverage competition to improve adaptive therapy in metastatic prostate cancer” aimed to present a calibrated and validated PCa stem cell model to test RBAT.

- Title

The author should be encouraged to reduce the number of title words. It is too long. May be a valuable focus in the main idea of the article. The “Range-bounded adaptive therapy” and “improve adaptive therapy in metastatic prostate cancer”.

- Abstract

The abstract should be revised and shortened to highlight the main results of the article.

A clear and concise objective of the study should be presented.

Line 21: Prostate cancer is a worldwide issue. Please revise.

- Introduction

A clear and concise objective of the study should be presented.

- Materials and Methods

Line 100-101: The authors should provide the clinical data collection period.

Author Response

We thank Reviewer 1 for reviewing our manuscript revisions. We greatly appreciate all of the comments and suggestions. Please find our point-by-point response to each of the concerns below.

Reviewer 1 Suggestions/Comments

- Title

The author should be encouraged to reduce the number of title words. It is too long. May be a valuable focus in the main idea of the article. The “Range-bounded adaptive therapy” and “improve adaptive therapy in metastatic prostate cancer”.

Response: Thank you for this suggestion. The title has been revised accordingly.

- Abstract

The abstract should be revised and shortened to highlight the main results of the article.

A clear and concise objective of the study should be presented.

Response: Thank you. We have revised the abstract to reflect the aims, methods, results, and conclusions of the study.

Line 21: Prostate cancer is a worldwide issue. Please revise.

 Response: The sentence has been revised accordingly (page 2, lines 66-67).

- Introduction

A clear and concise objective of the study should be presented.

 Response: Thank you. The objective of the study is outlined on page 3, lines 119-121. It states, “In this study, we will use our simple model to investigate how the proposed modifications from Hansen and Read could be used to enhance adaptive therapy in individual patients.”

- Materials and Methods

Line 100-101: The authors should provide the clinical data collection period.

Response: Thank you for this suggestion. The clinical trial design and results have been previously published. We have revised the text to include the appropriate citations for the interested reader (page 3, lines 134-135).

Reviewer 2 Report

Thank you for the opportunity to review the manuscript entitled, " Range-bounded adaptive therapy to further leverage competition to improve adaptive therapy in metastatic prostate cancer." The clinical topic is important. However, I have several comments to improve the quality of the manuscript.

1.    In the simple summary, please define adaptive therapy before going into benefits. It is unclear what it is from just reading the simple summary.

2.    The abstract can be substantially improved. It would benefit the paper if the authors added quantitative information. Moreover, certain things can be left out (e.g., prevalence of prostate cancer). The first half of the abstract can be condensed to 1-2 sentences so we can focus on the aims, methods, results, and conclusions.

3.    Could the authors include information on when data was collected in the abstract? Which institutions?

4.    It would benefit the paper if the authors included more than a single limitation in the study. One important one is the small power / study sample of just 16 patients.

5.    Could the authors include information on prevalence and/or incidence in the beginning of the introduction?

6.    Can you find a citation that shows that PSA progression can be defined as PSA increasing 25% or greater and at least 2 ng/mL above nadir, confirmed by a second measurement three or more weeks later. Alternatively, why was this specific threshold used?

7.    I am missing tables. E.g., a table 1 on patient characteristics?

Author Response

We thank Reviewer 2 for reviewing our manuscript revisions. We greatly appreciate all of the comments and suggestions. Please find our point-by-point response to each of the concerns below.

Reviewer 2 Suggestions/Comments

Thank you for the opportunity to review the manuscript entitled, " Range-bounded adaptive therapy to further leverage competition to improve adaptive therapy in metastatic prostate cancer." The clinical topic is important. However, I have several comments to improve the quality of the manuscript.

  1. In the simple summary, please define adaptive therapy before going into benefits. It is unclear what it is from just reading the simple summary.

Response: Thank you. The simple summary has been revised to include the definition of adaptive therapy (page 1, lines 11-12).

  1. The abstract can be substantially improved. It would benefit the paper if the authors added quantitative information. Moreover, certain things can be left out (e.g., prevalence of prostate cancer). The first half of the abstract can be condensed to 1-2 sentences so we can focus on the aims, methods, results, and conclusions.

Response: Thank you for this suggestion. The abstract has been revised to include quantitative information as well as focus on the aims, methods, results, and conclusions of the study.

  1. Could the authors include information on when data was collected in the abstract? Which institutions?

Response: The abstract has been revised to include the institution where the data was collected. As the clinical trial design and results have been previously published, we have revised the text to include the appropriate citations to find the data collection information for the interested reader (page 3, lines 134-135).

  1. It would benefit the paper if the authors included more than a single limitation in the study. One important one is the small power / study sample of just 16 patients. 

Response: Thank you for this suggestion. The discussion has been revised to include this limitation. We have also discussed two additional limitations, namely 1) waiting for the PSA to rise above  before commencing treatment, as PSA can rise sharply over a short period of time (page 6, lines 248-260), and 2) the initial waiting period before beginning treatment, which can result in a more than three-fold increase in an already large tumor burden depending on the patient (page 7, lines 261-266).

  1. Could the authors include information on prevalence and/or incidence in the beginning of the introduction?

Response: Thank you. The introduction has been revised to include this information (page 2, lines 66-67).

  1. Can you find a citation that shows that PSA progression can be defined as PSA increasing 25% or greater and at least 2 ng/mL above nadir, confirmed by a second measurement three or more weeks later. Alternatively, why was this specific threshold used?

Response: As these patients were metastatic, radiographic progression was used to determine clinical progression. However, the oncologists noted that a patient developed PSA progression once his PSA increased by more than 25% from one measurement to the next. A similar threshold was previously used in a clinical trial in metastatic hormone-sensitive prostate cancer patients (Sweeney et al. NEJM, 2015).

  1. I am missing tables. E.g., a table 1 on patient characteristics?

Response: Thank you for this suggestion. The clinical trial design, including patient demographics, have been previously published. We have revised the text to include the appropriate citations for the interested reader (page 3, lines 134-135).

Round 2

Reviewer 2 Report

The authors have done a nice job responding to my comments. The revised paper is much easier for me to follow. I do not have additional comments.